# Molecular and Physiological Functions of PACAP in Sweat Secretion

**DOI:** 10.3390/ijms24054572

**Published:** 2023-02-26

**Authors:** Michio Yamashita, Junko Shibato, Randeep Rakwal, Naoko Nonaka, Takahiro Hirabayashi, Brian J. Harvey, Seiji Shioda, Fumiko Takenoya

**Affiliations:** 1Department of Sport Sciences, Hoshi University School of Pharmacy and Pharmaceutical Sciences, 2-4-41 Ebara, Shinagawa-ku, Tokyo 142-8501, Japan; 2Department of Functional Morphology, Shonan University of Medical Sciences, 16-48 Kamishinano, Totsuka-ku, Yokohama, Kanagawa 244-0806, Japan; 3Institute of Health and Sport Sciences, University of Tsukuba, 1-1-1 Tennodai, Tsukuba, Ibaraki 305-8574, Japan; 4Department of Oral Anatomy and Developmental Biology, Showa University School of Dentistry, Shinagawa-ku, Tokyo 142-8555, Japan; 5Department of Molecular Medicine, Royal College of Surgeons in Ireland, Beaumont Hospital, Dublin D02 YN77, Ireland

**Keywords:** sweat gland, eccrine gland, PACAP, aquaporin5 (AQP5), DNA microarray

## Abstract

Sweat plays a critical role in human body, including thermoregulation and the maintenance of the skin environment and health. Hyperhidrosis and anhidrosis are caused by abnormalities in sweat secretion, resulting in severe skin conditions (pruritus and erythema). Bioactive peptide and pituitary adenylate cyclase-activating polypeptide (PACAP) was isolated and identified to activate adenylate cyclase in pituitary cells. Recently, it was reported that PACAP increases sweat secretion via PAC1R in mice and promotes the translocation of AQP5 to the cell membrane through increasing intracellular [Ca^2+^] via PAC1R in NCL-SG3 cells. However, intracellular signaling mechanisms by PACAP are poorly clarified. Here, we used PAC1R knockout (KO) mice and wild-type (WT) mice to observe changes in AQP5 localization and gene expression in sweat glands by PACAP treatment. Immunohistochemistry revealed that PACAP promoted the translocation of AQP5 to the lumen side in the eccrine gland via PAC1R. Furthermore, PACAP up-regulated the expression of genes (*Ptgs2*, *Kcnn2*, *Cacna1s*) involved in sweat secretion in WT mice. Moreover, PACAP treatment was found to down-regulate the *Chrna1* gene expression in PAC1R KO mice. These genes were found to be involved in multiple pathways related to sweating. Our data provide a solid basis for future research initiatives in order to develop new therapies to treat sweating disorders.

## 1. Introduction

Sweat is an exocrine fluid secreted by the skin and is an essential body fluid for thermoregulation and skin water regulation [1]. It is secreted from the sweat gland of the eccrine gland and apocrine gland, and 90% of the sweat secretory function is performed by the eccrine gland, of which 99% is water and the rest is inorganic. Sweat secretion is promoted by an increased body temperature and stress, etc., and contributes to the maintenance of homeostasis in the organism. It is known that abnormalities in sweat secretion can lead to dyshidrosis, such as hyperhidrosis, which causes excessive sweating, and anhidrosis, which prevents sweat secretion. It is known that dyshidrosis can result in not only a high fever and heat stroke but also causes serious skin conditions such as pruritus and erythema, which markedly reduce the quality of life of patients [2,3]. The cause of these disorders is thought to be one of the causes of nervous system diseases and other underlying diseases, such as autonomic dysrhythmia, but its basis is yet to be clarified [1]. Therapies such as using an anticholinergic drug and surgical treatment are available for the treatment of hyperhidrosis; however, no well-defined therapy exists for anhidrosis [4,5,6,7,8]. Previous studies have reported the involvement of neurotransmitters and peptides, such as acetyl choline (ACh), noradrenaline (NA) and non-noradrenergic transmitter (NANC), and atrial natriuretic peptide, in the regulation of sweat secretion in sweat glands [9]. It is known that ACh and NA, which are the typical neurotransmitters involved in sweat secretory action, act on the G protein-coupled receptor of the Gs and Gq system expressed in sweat gland cells and increase Ca^2+^ and cyclic adenosine monophosphate (cAMP) in cells. Increased levels of intracellular Ca^2+^ and cAMP are thought to promote water secretion by translocating aquaporin5 (AQP5), the main water channels in sweat glands, from the cytosol to the plasma membrane [10].

Pituitary adenylate cyclase-activating polypeptide (PACAP) is a bioactive peptide isolated and identified to potently activate the adenylate cyclase in pituitary cells from the hypothalamus of the sheep brain [11,12]. PACAP is composed of 27 or 38 amino acid residues, with PACAP27 and PACAP38 almost showing the same action to increase cytosolic cAMP levels. PACAP belongs to the vasoactive intestinal peptide (VIP)/secretin/glucagon family. PACAP and VIP share three GPCR receptors: the PAC1 receptor (PAC1R) and VPAC1R and VPAC2R. In addition, various physiological roles of PACAP are becoming clear from the analysis of the tissue distribution of PACAP and its receptors and in vitro and in vivo physiological and pharmacological studies. PACAP is considered to act as a neurotransmitter/modulator and in addition to its hormonal action, PACAP has been reported to have many physiological effects, including the promotion of glucose-dependent insulin secretion in pancreatic islet B cells, the regulation of pain inhibition, immune suppression, protection against ischemic neuronal cell death, and nerve regeneration [13,14,15,16,17,18].

In recent years, research has also shown that PACAP can stimulate the secretion of exocrine glands in tissues such as the lacrimal gland, sweat gland, and salivary gland [19]. It was reported that PACAP gene knockout (KO) mice developed corneal damage similar to the dry eye [20]. In addition, the function of PACAP in promoting tear secretion in the lacrimal gland has been revealed, and its mechanism of action has been shown to involve the activation of the cAMP/protein kinase A (PKA) pathway by PACAP through its action on PAC1R in the lacrimal gland, leading to the phosphorylation of AQP5 and promoting the translocation of AQP5 from the cytosol to the cell membrane [20]. In sweat glands, immunohistochemical analysis using human and mouse skin tissue has shown that PACAP is present in nerve endings near eccrine glands and that PAC1R is expressed in the acinar cells of eccrine gland secretory cells [21]. The addition of PACAP to the plantar surface of the mice’s forepaws has been reported to increase sweat secretion in mice, and this effect is inhibited by an inhibitor of PAC1R [21]. In addition, it is reported that PACAP has an ability to increase the cytosolic Ca^2+^ concentration through PAC1R and to promote the transition of AQP5 to the cell membrane in NCL-SG3 cells, which are immortalized human eccrine gland cells [22]. However, the intracellular signaling mechanisms by PACAP are poorly defined [9]. Therefore, in this study, we used PAC1R gene KO mice (PAC1R KO) and wild-type (WT) mice to observe changes in AQP5 localization and gene expressions (DNA microarray, GSE223124) in sweat gland tissues by PACAP in order to identify the cytosolic cell signaling mechanism.

## 2. Results

### 2.1. Distribution of PAC1R in Eccrine Gland in WT Mice and PAC1R KO Mice

The distribution of PAC1R immunoreactivity was observed by using fluorescent light microscopy. In the WT mouse, PAC1R was localized in the acinus of the eccrine gland (Figure 1A,B). In the PAC1R KO mouse, PAC1R was not observed in the eccrine gland (Figure 1C,D).

### 2.2. AQP5-Positive Vesicles Present in the Secretory Cells of the Mouse Eccrine Gland

We used confocal laser microscopy to construct a 3D model of mouse eccrine gland tissue and observed the intracellular localization of AQP5-like immunoreactivity (LI) vesicles using immunostaining. As a result, the presence of AQP5 positive vesicles, about 50 nm in diameter, both on the cell membrane and in the cytoplasm of mouse exocrine gland acinar cells, were found (Figure 2).

### 2.3. Changes in the Intracellular Localization of AQP5 Following Intradermal Injection of PACAP

In previous studies, it was reported that AQP5-containing vesicles transiently migrate to the cell membrane in the eccrine gland following PACAP treatment [21]. Therefore, in this study, the intracellular localization of AQP5-LI was observed using immunostaining at 30 and 60 min after an intradermal injection of PACAP to the WT mice and PAC1R KO mice. In the WT mice group 30 min after PACAP treatment, the localization of AQP5-LI in the acinus was strongly observed in the apical cell membrane over the vehicle-treated WT mice (Figure 3A,B). However, after 60 min of PACAP treatment, the localization of AQP5-LI was not observed as intensely in the acinus (Figure 3C,D). In addition, in the PAC1R KO mice group, the transition of AQP5 to the lumen of the acinus was weakly observed by PACAP administration (Figure 3E,H).

### 2.4. Changes in Gene Expression Levels in WT Mice and PAC1R KO Mice after Intradermal Administration of PACAP at 30 and 60 Minutes

Based on previous results, it has been suggested that the activation of PAC1R by PACAP leads to the translocation of AQP5 from the cytoplasm to the apical side of the secretory cells of the mouse eccrine gland [22]. Therefore, in order to clarify the mechanism by which PACAP promotes sweating, we observed gene expression changes in WT mice and PAC1R KO mice after an intradermal injection of PACAP in the skin of the forefeet for 30 and 60 min using DNA microarray analysis. The results revealed changes in the expression of multiple genes at 30 min after PACAP treatment, of which 70 genes were up-regulated and 45 genes were down-regulated in the WT mice, while 190 up-regulated genes and 43 down-regulated genes were found in the PAC1R KO mice (Figure 4). Additionally, 197 genes were up-regulated and 76 genes were down-regulated in the WT mice, while 170 genes were up-regulated and 39 genes were down-regulated in the PAC1R KO mice at 60 min after PACAP treatment (Figure 4).

Several genes involved in sweat secretion showed changes in their expression levels: in WT mice, three genes involved in sweat secretion, Ptgs2, Kcnn2, and Cacna1s, were upregulated at 30 min after PACAP treatment; in PAC1R KO mice, the gene whose expression was decreased was Chrna1s at 30 min after PACAP treatment (Table 1).

### 2.5. Top 20 Genes Whose Expression Was Altered by PACAP in WT and PAC1R KO Mice

Figure 5 shows the top 20 genes with expression changes in WT and PAC1R KO mice at 30 and 60 min after PACAP administration, showing those with expression changes of 2.0-fold or more and 0.5-fold or less, respectively. Comparing WT and KO, genes involved in antimicrobial activity (yellow) increased more in KO, and 13 of the top 20 genes were related to antimicrobial activity, especially at 60 min post-treatment. Genes related to muscle contraction (gray) were increased in WT, whereas they were decreased in KO. Trub2, which was decreased at 30 min in the KO, is an oxidative phosphorylation-related gene (green) that is responsible for ATP biosynthesis in the mitochondrial inner membrane electron transport system in a phosphorylation reaction that is coupled to the oxidation of NADH. Many Trub2 gene probes have been detected, and the change in the decreased Trub2 expression seems significant. Other cytokines such as Il6, Cxcl9, and Cxcl5 were also frequently expressed but were confirmed in both the WT and KO.

## 3. Discussion

### 3.1. PACAP Promotes Plasma Membrane Translocation of AQP5 via PAC1R

Previous in vivo studies reported that local sweat secretion promotion through PAC1R is elicited by administering PACAP into the subcutaneous tissue of the mouse footpad under anesthesia [21]. In addition, it has been reported that PACAP-related peptide VIP promotes sweat secretion through the cAMP pathway [23]. In the in vitro studies using human eccrine gland immortalized cells (NCL-SG3 cells), we have reported that PACAP plays important roles in the translocation of AQP5 from cytoplasm to the cell membrane by increasing the intracellular Ca^2+^ concentration by PAC1R [22]. However, there are less data showing genome-wide changes in sweat glands after the administration of PACAP. In this study, we first observed changes in the intracellular localization of AQP5 followed by the gene expression in the sweat glands after an intradermal injection of PACAP. At first, we confirmed that the location of PAC1R was in the eccrine gland of the WT mouse. Simultaneously, in the PAC1R KO mouse, PAC1Rs were not observed in the eccrine gland.

Using confocal laser scanning microscopy, we identified many AQP5-positive vesicles in the cytoplasm of eccrine gland secretory cells by analyzing the mouse eccrine gland in three dimensions. AQP5-LI in the eccrine gland was observed in small vesicles in the cytoplasm of secretory cells [24]. AQP5 has been reported to promote water secretion by transitioning to the cell membrane in response to external stimuli [25]. After an intradermal injection of PACAP, the WT mice had more AQP5 localized on the luminal side of the eccrine gland compared to the control group. However, after 60 min of PACAP administration, the localization of AQP5-LI vesicles on the luminal side decreased and its immunoreactivity increased in the cytoplasm. In the group of PAC1R KO mice administered with PACAP, a weak AQP5-LI was found on the cell membrane in the acinus. Therefore, it is suggested that in the mouse eccrine gland, PACAP promotes the transition of AQP5 from cytoplasm to the cell membrane acting through PAC1R.

### 3.2. Results from the Whole Genome DNA Microarray Analysis

Figure 6 describes the gene categories for their involvement in the PACAP-induced membrane localization of AQP5.

#### 3.2.1. Expression of Genes Involved in Sweat Secretion

The expression levels of four genes (Ptgs2, Kcnn2, Cacna1s, and Chrna1) known to be involved in sweat secretion were found to be elevated in WT mice at 30 min. One of these genes, prostaglandin endoperoxide synthase 2 (Ptgs2/COX-2), has been suggested to be involved in sweat secretion rather than vasodilation through studies in which COX inhibitors and selective COX2 inhibitors were used in human subjects subjected to exercise. The mechanism of sweating mediated by COX is thought to work though the interaction with nitric oxide synthase (NOS), and it is suggested that these interactions may activate Cl^-^ channels and/or the Na^+^/K^+^-ATPase enzyme [26,27,28,29]. *Kcnn2* encodes the small conductance calcium-activated potassium channel 2 (SK); the SK channel activation is voltage independent and depends on intracellular calcium levels [30]. Kcnn4, which belongs to the same potassium–calcium-activated channel family, is one of the main regulatory factors of eccrine glands, and its expression has been reported to co-localize with Foxa1 and eccrine gland secretory cells, and Kcnk5, a potassium voltage-gated channel, double knockout mice were found to have significantly reduced sweating compared to WT mice [27].

In particular, Cacna1s is a calcium voltage-gated channel and previous studies have reported that the inhibition of this channel leads to the suppression of sweat secretion [31]. Changes in the intracellular calcium concentrations within the eccrine glands have been suggested to be involved in a calcium influx from both intracellular and extracellular sources. Furthermore, an increase in the intracellular calcium concentration has been shown to promote sweating through the action of calcium-dependent chloride channels (CaCCs) and other factors [32,33]. Previous research has also demonstrated that PACAP can increase intracellular calcium concentrations in eccrine gland cells and promote the translocation of AQP5 to the cell membrane [22,34].

The decreased expression of the cholinergic receptor nicotinic α1 subunit (Chrna1), which plays an important role in sweating in eccrine glands, was observed in PAC1R KO mice in this study. CHRNA1 has been reported as a causative factor in primary focal hyperhidrosis and, similarly to CACNA1, it is strongly involved in the mechanism of eccrine sweat production in mice injected with the siRNA Chrna1 gene [31].

#### 3.2.2. Expression of Genes Involved in Inflammation

Aquaporins function as inflammatory mediators in some lesions. The dysfunction of aquaporins is involved in the development of inflammatory skin diseases characterized by the disruption of the skin barrier [35,36,37]. Genes involved in inflammation, such as neutrophil degranulation and antibacterial activity, were increased in KO mice compared to WT mice. In particular, S100a8 and S100a9 have been shown to be increased in inflammatory skin diseases [38,39,40], suggesting that the PAC1R KO mice have skin inflammation due to defective aquaporin action. This is likely because the genes KRT17 (2.1-fold change), KRT16, and KRT6A (2.3-fold change), whose expression increased at 60 min in KO mice, are upregulated during skin damage and inflammation. Recent studies have recognized that keratinocytes (KCs) located at the skin surface exposed to external stimuli, such as pathogens and/or injury, are activates that in turn secrete an array of alarmin molecules, and which can be considered a rapid and specific innate immune response [41].

#### 3.2.3. Expression of Genes Involved in Contractile Proteins

Myosin light chain phosphorylation in the lung plays an important role in cell contraction, extracellular permeability, and lung water homeostasis, and the inhibition of myosin light chain kinase leads to a decreased AQP5 expression. Contraction induced by myosin light chain phosphorylation can lead to tight junction opening because the cell junction complex is bound to actin and non-muscle myosin in the cytoskeleton [42]. Furthermore, the cytosolic Ca^2+^ concentration has been shown to upregulate the AQP5 expression by regulating it via motor proteins such as myosin and dynein, resulting in AQP5 plasma membrane redistribution. Tmod1, whose expression was decreased in KO mice, has been reported to be involved in the regulation of the body water balance [43]. In the kidney, actin and actin-related proteins are involved in the regulation of water and salt homeostasis. Several channels interact with actin, including aquaporin 2 (AQP2), the cystic fibrosis transmembrane regulator (CFTR), and the epithelial sodium channel (ENaC).

#### 3.2.4. Expression of Genes Involved in Ca^2+^·ATP

Increases in intracellular calcium and ATP are involved in the translocation of AQP5 [44,45]. Panx3 is a member of the pannexin family found in skin and is present at the plasma membrane and endoplasmic reticulum membrane and functions as a channel to move ATP and Ca^2+^ into and out of the cell. Pannexin 3 channels are also involved in skin homeostasis and wound healing [46,47]. Trub2, whose expression was decreased in KO mice, is involved in oxidative phosphorylation, and mitochondrial oxidative phosphorylation is a major pathway for ATP generation [48].

#### 3.2.5. Expression of Genes Involved in Blood Coagulation

Disseminated intravascular coagulation syndrome (DIC) is an acquired syndrome characterized by the extensive activation of coagulation, leading to the dysfunction of multiple organs in the body [49]. Coagulation reactions have also been identified in Sjögren’s syndrome [50].

#### 3.2.6. The TBC1D1 Gene

Although there are no reports of an association with AQP5, TBC1D1 (2.5-fold change), which is involved in an enhanced glucose uptake via glucose transporter 4 (GLUT4) translocation, was identified at 30 min in the WT mice. TBC1D1 is involved in the regulation of glucose processing and the substrate metabolism within the skeletal muscle and is essential for the insulin secretory function. An important role for TBC1D1 in the translocation of GLUT4 to the skeletal muscle protoplasm membrane via exercise, and contraction has been reported [51,52]. Recently, it has been reported that TBC1D1 is potentially associated with the etiology of atopic dermatitis in dogs, a chronic inflammatory and pruritic skin disease [53]. Therefore, we are interested in whether TBC1D1 is involved in the translocation of AQP5 by PACAP.

Overall, from this study, it is suggested that the sweat-promoting effects of PACAP mediated by PAC1R are due to its ability to promote the translocation of AQP5 to the luminal side of the eccrine gland. Additionally, various pathways via PAC1R were identified in mouse skin, including those involved in sweat secretion that was affected by an intradermal injection of PACAP. In the future, it will be necessary to deeply investigate how these pathways contribute to sweat secretion by using the inhibitors of these pathways and the eccrine cells obtained from the study.

## 4. Material and Methods

### 4.1. Animals

All experimental procedures involving animals were approved by the Institutional Animal Care and Use Committee of Hoshi University. The male Adcyap1r1 -/- mice (PAC1R KO) and the male C57BL/6 (WT) were bred and maintained under specific pathogen-free conditions in the animal facility of Hoshi University. Animals were housed with a 12 h/12 h light/dark cycle and were provided free access to water and standard rodent chow. Nine- to ten-week-old mice were used for the experiments.

### 4.2. Administration of PACAP to the Plantar Surface of Mice’s Forepaws

Adult C57BL/6 mice (Tokyo Laboratory Animals Science Co., Ltd., Tokyo, Japan) and the PAC1R KO mice were anaesthetized with three types of mixed anesthetic agents (5 mL kg^−1^, i.p.). These anesthetic agents were midazolam (0.3 mg kg^−1^, Maruishi Seiyaku, Osaka, Japan), medetomidine (4 mg kg^−1^, Kyoritsu Seiyaku, Tokyo, Japan), and butorphanol (5 mg kg^−1^, Meiji Seika, Tokyo, Japan). In total, 5 µL of the vehicle (saline containing 0.1% BSA) containing 10^−7^ mol/L of PACAP and 5 µL of the vehicle alone were intradermal injected into the center of the footpad with a 26G needle attached to a 10 µL glass syringe (HAMILTON, Reno, NV, USA) [54].

### 4.3. Histology

The mice were euthanized 30 and 60 min after the administration of PACAP by spinal dislocation and the skin from the forepaws was harvested. The harvested samples were fixed overnight at 4 °C in a 4% paraformaldehyde solution in 0.1 M of phosphate buffer (PB). The fixed tissues were processed in 20% sucrose solution overnight and 30% sucrose solution for 48 h, followed by embedding. Cryosections (10 μm thickness) were cut from the frozen tissue using a cryostat and used for immunostaining.

### 4.4. Determination of the Location of PAC1R in Mouse Skin

The cryosections were washed with phosphate-buffered saline (PBS) and then blocked with 5% horse serum in PBS for 1 h and incubated overnight at 4 °C in a solution of the following primary antibody: rabbit anti-PAC1R antibody (1:200, developed by our laboratory) [55]. It was then washed with PBS and reacted with a secondary antibody: Alexa Flour 594 Donkey anti-rabbit IgG (1:800, Invitrogen, MA, USA) for 60 min at room temperature. It was re-washed with PBS, followed by nuclear staining with DAPI (1:10,000, Invitrogen) for 3 min at room temperature; then, it was washed with PBS and encapsulated using a VECTASHIELD Vibrance Antifade Mounting Medium (Vector Laboratories, Newark, CA, USA). After drying, the samples were visualized and imaged on a fluorescent microscope (BZ-X710; Keyence, Osaka, Japan).

### 4.5. Confirming the Intracellular Location of AQP5 in Sweat Gland

The cryosections were washed with PBS and then blocked with 5% horse serum in PBS for 1 h and incubated overnight at 4 °C in a solution of the following primary antibody: rabbit anti-AQP5 antibody (1:200, Merck Millipore, Billerica, MA, USA). It was then washed with PBS and reacted with a secondary antibody, Alexa Flour 488 Donkey anti-rabbit IgG (1:800, Invitrogen), for 60 min at room temperature. It was re-washed with PBS followed by nuclear staining with DAPI (1:10,000, Invitrogen) for 3 min at room temperature; then, it was washed with PBS and encapsulated using a VECTASHIELD Vibrance Antifade Mounting Medium (Vector Laboratories). After drying, the stained sections were photographed cross-sectionally using a confocal laser microscope A1R/Ti-2E (Nikon, Tokyo, Japan).

### 4.6. Whole Genome DNA Microarray Analysis

After an intradermal injection of PACAP to six WT mice and PAC1R KO mice each, the skin of the forepaws was sampled at 30 and 60 min later, which were immersed immediately post-dissection in liquid nitrogen and thereafter transferred to an −80 °C deep freezer. The samples were individually ground in liquid nitrogen to prepare very fine powders for subsequent downstream gene expression analyses [56,57]. Prior to each DNA microarray analysis, the total RNA was extracted using an optimized protocol using the QIAGEN RNeasy Mini Kit (QIAGEN, Germantown, MD, USA). Optimization was done based on our expertise in extracting total RNA from multiple samples each requiring an individual trial experiment/s to optimize the protocol for each of the organisms, tissues, and cells in aspects of the amount of tissue and time of grinding, the amount of phenol (including preparation of the reagents), the use of a single or double chloroform extraction, ethanol washes, etc., and finally confirming both the quality and quantity of total RNA, and, as also mentioned below, using RT-PCR to check the gene expressions as a positive control. The quantity and quality were measured spectrophotometrically with DeNovix DS-11 (DeNovix, Wilmington, DE, USA) and re-confirmed using formaldehyde-agarose gel electrophoresis [56,57]. Briefly, the obtained total RNA quality measurements revealed a good quality, as demonstrated by the A260/280 values > 1.8, A260/230 > 1.8; in our experiments, we always aim for a higher value of more than 2.0 to 2.3. Furthermore, the total RNA was obtained in an optimum quantity of >300 ng/μL. Following the visualization for confirming the pre-experimental integrity of the total RNA subunits by gel electrophoresis, an additional step comprised of cDNA synthesis was used for examining the quality of the synthesized cDNA by determining the expression of a commonly used housekeeping gene, *glyceraldehyde 3-phosphate dehydrogenase* (*GAPDH*). This was done by RT-PCR (AffinityScript QPCR cDNA Synthesis Kit; Strategene, Agilent Technologies, Santa Clara, CA, USA, and an Emerald Amp PCR Master Mix; TaKaRa, Shiga, Japan) on a Thermal Cycler (Applied Biosystems, Tokyo, Japan). Moreover, only gene-specific primers were used, whose sequences are shown in the articles. Post-electrophoresis, the PCR products were visualized and quantified using a ChemiDoc XRS + imaging system (Bio-Rad Laboratories, Hercules, CA, USA). All the steps followed in the sequence provided confidence that the best quality total RNA (sample) was used in the DNA microarray chip. As a last step, the Agilent mouse, whole genome was 4 × 44 K (G4122F) DNA slide (composed of 4 chips on one slide), was used for the microarray analysis, which was performed according to the manufacturer’s instructions (Agilent Technologies, Santa Clara, CA, USA) and detailed in our publications [56,57]. Briefly, the total RNA (400 ng) was labeled with either Cy3 or Cy5 dye using an Agilent Low RNA Input Fluorescent Linear Amplification Kit. Fluorescently labeled targets of the control as well as the treated samples were hybridized to the same microarray slide with 60 mer probes. A flip labeling (dye-swap or reverse labeling with Cy3 and Cy5 dyes) procedure was followed to nullify the dye bias associated with an unequal incorporation of the two Cy dyes into cDNA. Hybridization and wash processes were performed according to the manufacturer’s instructions, and hybridized microarrays were scanned using an Agilent Microarray scanner G2505C. For the detection of significantly differentially expressed genes between the control and treated samples, each slide image was processed by Agilent Feature Extraction software (version 9.5.3.1). This program measures the Cy3 and Cy5 signal intensities of whole probes. Dye-bias tends to be signal intensity-dependent; therefore, the software selected probes using a set by a rank consistency filter for dye normalization. Said normalization was performed by LOWESS (locally weighted linear regression) which calculates the log ratio of dye-normalized Cy3 and Cy5 signals, as well as the final error of the log ratio. The significance (P) value was based on the propagate error and universal error models. In this analysis, the threshold of significant differentially expressed genes was <0.01 (for the confidence that the feature was not differentially expressed). In addition, erroneous data generated due to artifacts were eliminated before the data analysis using the software. The whole-genome DNA microarray data of treated skin (forepaws) of C57BL/6 WT and PAC1R KO mice have been submitted to NCBI’s GeneExpression Omnibus under the GEO series accession number GSE223124 (https://www.ncbi.nlm.nih.gov/geo/query/acc.cgi?acc=GSE223124; accessed on 19 February 2023).

## Figures and Tables

**Figure 1 ijms-24-04572-f001:**
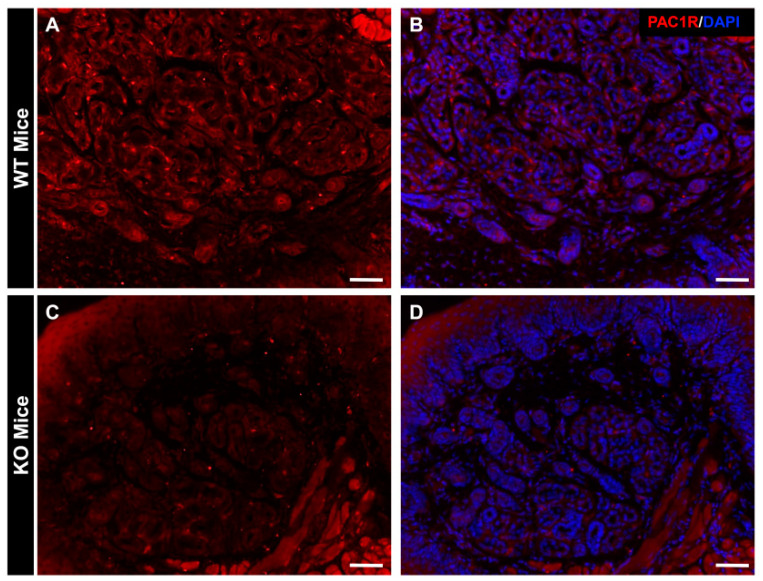
Analysis of PAC1R distribution in mouse eccrine gland. The WT mouse and PAC1R KO mouse skins were immunostained with PAC1R (red; **A** and **C**) and counter stained with DAPI (nuclei, **B** and **D**). Scale bar = 50 µm.

**Figure 2 ijms-24-04572-f002:**
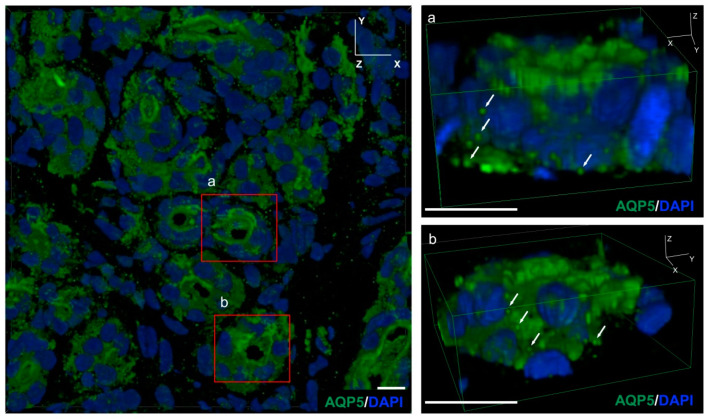
A 3D analysis of the AQP5 vesicles localization in mouse eccrine cells. Mouse skins were immunostained with AQP5 (green) and counter stained with DAPI (nuclei). (**a**,**b**): The white arrows highlight AQP5-immunoreactive vesicles in the cytoplasm of mouse skin. Scale bar = 10 µm.

**Figure 3 ijms-24-04572-f003:**
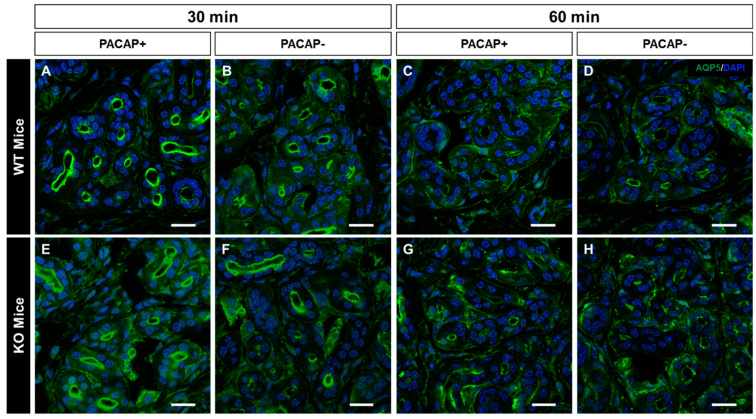
Changes in the intracellular localization of AQP5 in WT mice and KO mice after PACAP addition. Mouse skins were immunostained with AQP5 (green) and counter stained with DAPI (nuclei). Representative confocal micrographs of mouse eccrine gland transfected with (**A**) 30 min after PACAP treatment, (**B**) 30 min after vehicle treatment, (**C**) 60 min after PACAP treatment, (**D**) and 60 min after vehicle treatment in WT mice. Representative confocal micrographs of mouse eccrine gland transfected with (**E**) 30 min after PACAP treatment, (**F**) 30 min after vehicle treatment, (**G**) 60 min after PACAP treatment, (**H**) and 60 min after vehicle treatment in PAC1R KO mice. Scale bar = 10 µm.

**Figure 4 ijms-24-04572-f004:**
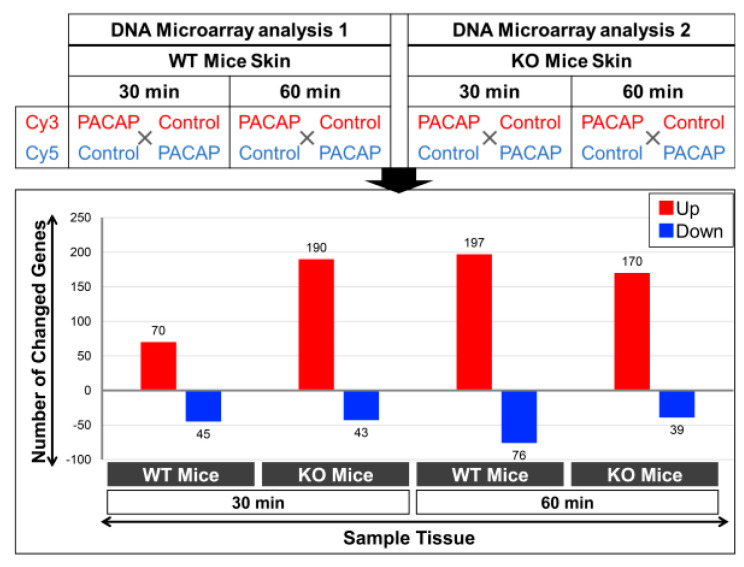
Differentially expressed genes in the WT mice and KO mice after the addition of PACAP at 30 and 60 min by DNA microarray analysis. Two DNA microarray chips were used for each sample condition, i.e., a dye-swap approach was used to label one set of RNAs by Cy3 and the other by Cy5 for each control and PACAP sample, respectively, as detailed in the Materials and Methods section. A 4 × 44-K format slide containing 4 chips is illustrated for the current study. The histograms show the up (red)- and down (blue)-regulated gene (≧/≦ 2.0/0.5-fold) numbers in each examined tissue. The gene expression data are publicly available under the GEO series accession number GSE223124 (https://www.ncbi.nlm.nih.gov/geo/query/acc.cgi?acc=GSE223124; accessed on 19 February 2023).

**Figure 5 ijms-24-04572-f005:**
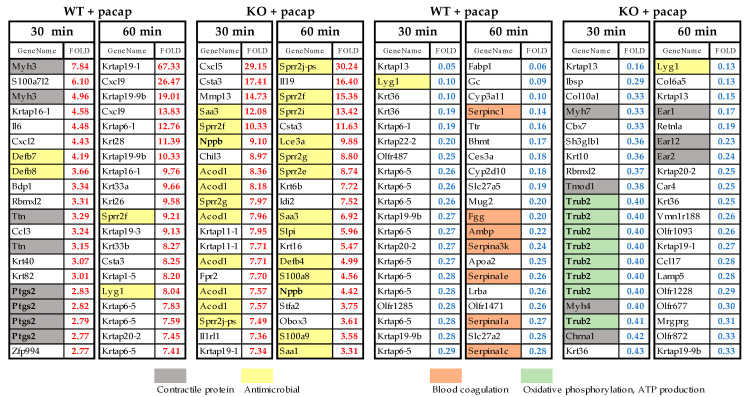
Top 20 genes expressions that were altered by PACAP in WT mice and PAC1R KO mice. Genes involved in contractile protein-related genes (gray), antimicrobial activity (yellow), blood coagulation-related genes (orange), and oxidative phosphorylation- and ATP production-related genes (green). The gene expression data are publicly available under the GEO series accession number GSE223124 (https://www.ncbi.nlm.nih.gov/geo/query/acc.cgi?acc=GSE223124; accessed on 19 February 2023).

**Figure 6 ijms-24-04572-f006:**
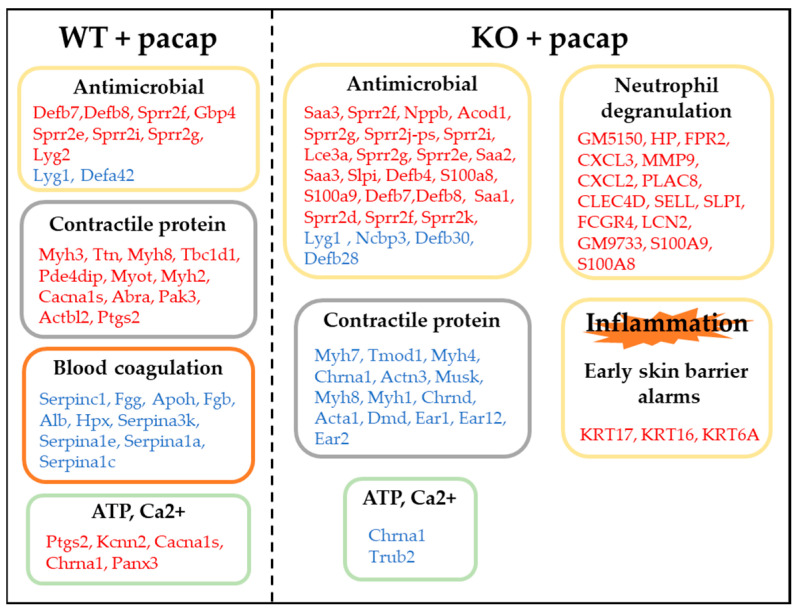
Summary of the genes that may be involved in PACAP-induced membrane translocation of AQP5. Red, up-regulated genes; and blue, down-regulated genes. The gene expression data are publicly available under the GEO series accession number GSE223124 (https://www.ncbi.nlm.nih.gov/geo/query/acc.cgi?acc=GSE223124; accessed on 19 February 2023).

**Table 1 ijms-24-04572-t001:** Genes involved in sweat secretion that changed upon PACAP addition. The gene expression data are publicly available under the GEO series accession number GSE223124 (https://www.ncbi.nlm.nih.gov/geo/query/acc.cgi?acc=GSE223124; accessed on 19 February 2023).

Gene Name	Fold			
	30 min		60 min	
	WT	KO	WT	KO
*Ptgs2*	2.83	1.37	1.25	1.84
*Kcnn2*	2.60	0.68	1.18	0.96
*Cacna1s*	2.22	0.58	0.95	1.20
*Chrna1*	1.93	0.42	1.28	1.07

## Data Availability

All the whole-genome DNA microarray data are publically available for the scientific community at the NCBI’s GeneExpression Omnibus under the GEO series accession number GSE223124.

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
