# Peer review of "Molecular and Physiological Functions of PACAP in Sweat Secretion"

_ijms, 2023, doi:10.3390/ijms24054572_

Round 1

Reviewer 1 Report

PACAP plays a role in regulating sweat gland function. Studies have shown that PACAP stimulation increases sweat secretion in rats and mice, suggesting that it may play a role in regulating sweat gland function. PACAP has been implicated in the regulation of thermoregulation and body temperature and by modulating sweat secretion, PACAP may play a role in regulating body temperature and preventing overheating during physical activity. Recent research has shown that PACAP promotes the translocation of AQP5 to the cell membrane and up-regulates gene expression involved in sweat secretion in mice. In this study, the localization and expression of AQP5 in sweat glands were evaluated by comparing Adcyap1r1 -/- mice (PAC1R 281 KO) and C57BL/6 (WT) mice. PACAP has been shown to promote the translocation of AQP5 to the cell membrane in the eccrine gland via PAC1R and up-regulate gene expression (Ptgs2, Kcnn2, Cacna1s) involved in sweat secretion in wild-type mice. PACAP treatment also down-regulated the gene expression of Chrna1 in PAC1R knockout mice. These genes are involved in multiple pathways related to sweating.

The study indicates that PACAP-mediated promotion of sweat secretion is caused by its influence on the movement of AQP5 in the eccrine gland. The intradermal injection of PACAP also affects various pathways involved in sweat secretion in mouse skin. These findings provide a basis for future research initiatives to develop new therapies to treat sweating disorders.

The topic fits to the journal scope, the questions are well defined. The manuscript is clear, relevant to the field, and presented in an organized manner. The cited 51 references are mostly stale. However, they are relevant and do not contain an excessive amount of self-referencing. Of the 51 references, only 8 are more than 5 years old, this ratio needs improvement and more recent publications have to be included. The scientific validity and experimental design are suitable, one can repeat the experiments with the data provided. The results are reproducible based on the information in the methods section. The four figures are appropriate. The conclusions are consistent with the evidence and arguments presented.

line 106 “in a 3D approach” is not necessary.

line 107 I suggest, “the presence of AQP5 positive vesicles, about 50 nm in diameter, both on the cell membrane and in the cytoplasm of mouse exocrine gland acinar cells”

line 125 fig 3. An unnecessary red scale bar presents on most slides.

line 146 I suggest to rephrase this sentence. it can be something like this: Some genes involved in sweat secretion showed changes in expression levels.

line 203 Please, rephrase the sentence, I suggest: Additionally, we noted a change in the gene expression levels after PACAP treatment.

line 287 Please, include the agents for anaesthesia.

line 292 In which way the animals were euthanised?

line 296 Followed by embedding and freezing.

line 307 Fluorescent microscope

Q1 Why did you choose not to include the data from 0 minutes? A comparison of WT vs KO gene expression data would be interesting in itself. Including the basic expression levels would provide more informative results. This finding also holds true for IHC.

Q2 Can you please specify the sex of the animals used in the experiments? Please include this information in the manuscript. Do you think it has the potential to affect the outcomes of the experiments?

Q3 I would be interested whether an official animal experiment permit is required in your country to perform the experiment or a permission from the University’s Ethical Committee is sufficient? (anesthesia and drug administration)

Q4 May I ask how the protocol for RNA extraction using the Quiagen kit was optimized?

Reviewer 2 Report

The authors aimed to describe the molecular mechanism of the induction of sweat secretion mediated by PACAP.

The subject is interesting, however, the manuscript suffers from serious drawbacks that do not allow its publication in the present form.

It is not entirely clear why authors use the term AQP5-like immunoreactivity. This raises the question whether the authors are not confident that the observed signal originates from AQP5 protein.

Fig 2: The authors state that AQP5 is located in the vesicles in the cytosol. However, there is a strong green signal in cell cytoplasm. The authors should explain the reason for such signal. Is this an imaging artefact or true signal for cytoplasmic AQP5?

Fig3: The authors claim that while AQP5 is localizing in the apical membrane of sweat glands in wt mice this is not the case in the KO mice. However, the data in the figure does not support this statement as a transient increase in the green signal can be observed also in the KO mice. In addition, to back up their claims the authors should provide high-quality images in much higher resolution. 

Fig4 and the analysis of microarray data: The manner how data is presented does not contribute to our understanding of the intracellular signaling pathways leading to the induction of AQP5 translocation and sweat secretion. A differential analysis of gene expression between wt and KO mouse samples should be additionally performed and intracellular as well as extracellular signaling events outlined. 

Round 2

Reviewer 2 Report

All the criticisms are sufficiently addressed by the authors.